# Epidural Analgesia and Back Pain after Labor

**DOI:** 10.3390/medicina55070354

**Published:** 2019-07-09

**Authors:** Anastasija Malevic, Dalius Jatuzis, Virginija Paliulyte

**Affiliations:** 1Clinic of Infectious Diseases and Dermatovenerology, Vilnius University Faculty of Medicine Institute of Clinical Medicine, Vilnius University Hospital Santaros Klinikos, J. Kairiūkscio 2, LT-08411 Vilnius, Lithuania; 2Clinic of Neurology and Neurosurgery, Vilnius University Faculty of Medicine Institute of Clinical Medicine, Vilnius University Hospital Santaros Klinikos, Santariskiu 2, LT-08661 Vilnius, Lithuania; 3Clinic of Obstetrics and Gynecology, Vilnius University Faculty of Medicine Institute of Clinical Medicine, Vilnius University Hospital Santaros Klinikos, Santariskiu 2, LT-08661 Vilnius, Lithuania; 4Clinic of Obstetrics and Gynecology of Vilnius University, Centre of Obstetrics and Gynecology, Vilnius University Hospital Santaros Klinikos, Santariskiu 2, LT-08661 Vilnius, Lithuania

**Keywords:** back pain, post-partum, epidural analgesia

## Abstract

*Background and Objectives*: The aim of this survey was to assess the impact of epidural analgesia on post-partum back pain in post-partum women. *Materials and Methods*: The questionnaire was completed by post-partum women during the first days after delivery. Six months later, the women were surveyed again. The response rate was 70.66%, a total of 212 cases were included in the statistical analysis. The statistical analysis of the data was conducted using SPSS^®^*Results*. Seventy-nine (37.26%) women received epidural analgesia, 87 (41.04%) intravenous drugs, and 46 (21.7%) women gave birth without anesthesia. The prevalence of post-partum back pain was observed in 24 (30.38%) women of the epidural analgesia group, in 24 (27.58%) subjects of the intravenous anesthesia group, and in 14 (30.43%) women attributed to the group of subjects without anesthesia. The correlation between post-partum back pain and the type of anesthesia was not statistically significant (*p* = 0.907). Six months later, the prevalence of back pain was found in 31.65% of women belonging to the epidural analgesia group, in 28.74% of women with intravenous anesthesia, and in 23.91% of women without anesthesia. The correlation between complaints of back pain six months after delivery and the type of anesthesia applied was not statistically significant (*p* = 0.654). *Conclusions*. The labor pain relief technique did not trigger the increased risk of back pain in the early post-partum period and six months after delivery.

## 1. Introduction

Labor is among the most painful experiences in a woman’s life, therefore, management of childbirth pain is a crucial moment not only in providing women in labor with more comfort, but also in relieving their stress and suffering [1,2,3,4]. The technique selected from a wide range of available techniques is aimed to relieve pain and depends on the mode of delivery, personal choice, and doctor’s recommendations [5]. Epidural analgesia is an effective and widely used treatment for labor pain. Non-medical pain relief methods or intravenous opioid-analgesics can provide an alternative in situations where regional analgesia is contraindicated or if less invasive methods are preferred by the woman or doctors [3,4,6].

Despite the fact that many scholars refer to epidural anesthesia as a gold standard for pain control in obstetrics, the most frequent concern of our patients receiving epidural analgesia is post-partum back pain [7,8]. The data obtained from the available literature on epidural analgesia during delivery are still controversial. Previous studies have suggested that labor epidural analgesia might be associated with an increased incidence of back pain in the post-partum period. Taking into account that the initial studies were retrospective, however, the findings presented by prospective studies did not reveal the relationship between labor epidural analgesia and long-term post-partum back pain [8,9].

The aim of this survey was to assess the influence of epidural analgesia on post-partum back pain in post-partum women.

## 2. Materials and Methods

In 2016, a prospective continuous survey was conducted at the Center of Obstetrics and Gynecology of Vilnius University Hospital Santaros Klinikos. During the first post-partum days, women completed the anonymous questionnaire composed of general items (i.e., age, marital status, education, height (cm), final weight (kg), concomitant illnesses, diseases related to spinal pathology), items on obstetrics (i.e., number of pregnancies and deliveries, duration of labor, type of analgesia, newborn’s data), and special items (i.e., back pain after previous pregnancy, back pain in current pregnancy and after delivery, frequency and intensity (digital analog scale) of pain, pain relief techniques, accompanying symptoms (headache, shoulder/neck symptoms, urinary incontinence, tingling or weakness in the arms or legs), and the effect of pain on the woman’s daily activities and quality of sleep).

In all, 300 questionnaires were distributed. The study did not include women suffering from back pain before pregnancy due to the diagnosed spinal pathology or past history of back injuries (4.33%) and women after caesarean section.

Six months after delivery, the respondents were surveyed again by e-mail or telephone. The response rate was 73.87%. The study did not include women who did not report the results of a repeated survey six months after labor. In total, 212 cases were included in the analysis of the statistical data.

The study was approved by the Medical Ethics Commission of Vilnius University Hospital Santaros Clinic (No.GR-1088, 01-03-2016). The study was approved by the Medical Ethics Commission of Vilnius University Hospital Santaros Clinic (01-03-2016).

The primary outcome variable was back pain quantified by self-reports (yes/no), pain scores (digital analog scale), and the back-pain impact on daily activities and the quality of sleep (yes/no).

The data were processed using Microsoft Excel^®^. The statistical analysis of the data was conducted using SPSS^®^ 23.0 software. The average ± standard deviation was employed to evaluate both quantitative (normal distribution) and qualitative data-frequency/percentages. The relationship between the quantitative variables was evaluated by carrying out a one-factorial dispersion analysis (ANOVA test) with a confidence interval (95% CI). The value of *p* < 0.05 was considered statistically significant.

## 3. Results

Out of 212 women, 79 (37.26%) women received epidural analgesia, 87 (41.04%) intravenous drugs (sol. Phentanyli 0.005% 2 mL (SANITAS, Lithuania, Kaunas)), and 46 (21.7%) women gave birth without anesthesia. The average age of women participating in the survey was 29.76 (±5.56) years. The distribution of women by age is presented in Figure 1. Other demographic characteristics are shown in Table 1.

In terms of age, weight gain, and height, there were no statistically significant differences observed between anesthesia groups. Furthermore, the comparison of women’s education and marital status among anesthesia groups revealed no statistically significant difference.

Primiparous women comprised 47.17% (*n* = 100) of all the groups, while the remaining percentage were multiparous. There was a significant difference between the pain relief technique used by primiparous and multiparous (*p* = 0.002). Epidural analgesia was found to be more frequently chosen by primiparous (61.5%), while childbirth without anesthesia (78.3%) was more frequently observed in multiparous women. An even distribution was found in the group of women with intravenous anesthesia.

In all groups of women, the mean duration of labor was 473 ± 198.94 min (7 h 53 min). A significant difference between anesthesia groups was observed in the mean duration of labor (*p* = 0.001). The longest delivery was recorded in the group of women with epidural anesthesia (510.13 ± 180.61 min), while the shortest was in the group of women without anesthesia (374.61 ± 177.98 min). The mean duration of first and second periods of labor in the epidural analgesia group, compared to other groups, was significantly longer (*p* = 0.002; *p* < 0.001). There was no statistically significant relationship between the mean total duration of labor, mean duration of first and second periods of labor, and post-partum back pain (*p* = 0.104; *p* = 0.275; *p* = 0.156). The remaining obstetric characteristics of women are presented in Table 2.

The evaluation of the data of previous deliveries showed a statistically significant relationship between the pain relief technique selected in the past and pain relief techniques deployed for the current delivery (*p* < 0.001). Also, back pain after previous labor had an impact on the pain relief technique applied in current delivery (*p* = 0.009) (Table 3).

A few days after delivery, the prevalence of post-partum back pain was observed in 24 (30.38%) women of the epidural analgesia group, in 24 (27.58%) subjects of the intravenous anesthesia group, and in 14 (30.43%) women of the group without anesthesia. There was no statistically significant correlation observed between post-partum back pain and the type of anesthesia (*p* = 0.907). The majority of women complained of pain in the waist area (64.52%) and recurrent pain a few times a day (48.39%). The evaluation of the frequency of pain and the site of pain among the groups of women after anesthesia showed no statistically significant difference (*p* = 0.132 and *p* = 0.275, respectfully).

The pain scores (digital analog scale) ranged from 0 to 7. The general mean of pain scores was 3.26 ± 1.59. The 3.17 ± 1.24 mean of pain scores was found in the epidural analgesia group, 3.33 ± 1.96 in the intravenous anesthesia group, and 3.29 ± 1.64 in the group without anesthesia (Figure 2). The survey revealed that a few days after labor, the pain relief technique applied during delivery did not impact the intensity of post-partum back pain (*p* = 0.503).

Women were surveyed on accompanying symptoms that could have been reported alongside with back pain. A few days after childbirth, the frequency of back pain symptoms varied from 3.23% to 12.9% (Table 4). The comparison of the incidence of various accompanying symptoms among the anesthesia groups was not statistically significant.

Six months after delivery, the prevalence of back pain was observed in 25 (31.65%) women of the epidural analgesia group, in 25 (28.74%) subjects of the intravenous anesthesia group, and in 11 (23.91%) women of the group of subjects without anesthesia. Six months after delivery, the correlation between back pain and the type of anesthesia was not statistically significant (*p* = 0.654). Figure 3 shows the comparison of post-partum back pain observed a few days after delivery and six months after labor

Six months after delivery, the majority of women complained of pain arising in different sites of the back (55.74%) or in the waist area (32.79%), frequency of pain occurrence ranged from a few times per week to a few times per month (62.3%). The survey revealed that pain in the waist area was significantly more frequent in the group of women who gave birth without anesthesia (90.9%) and pain in different sites of the back in the group of women who gave childbirth with epidural analgesia or intravenous drugs. The evaluation of the frequency of pain occurrence (*p* = 0.367) revealed no statistically significant differences between anesthesia groups.

Six months after labor, the pain scores (digital analog scale) ranged from 1 to 6. The general mean of pain scores was 3.13 ± 1.28. The mean of pain scores in the epidural analgesia group was 3.36 ± 1.04, in the intravenous anesthesia group—3.12 ± 1.50, and in the group without anesthesia—2.64 ± 1.20. The higher intensity of back pain was observed in the epidural anesthetic group (*p* = 0.004).

Back pain affected the quality of sleep (*p* = 0.011) and daily activities (*p* = 0.004) in the group of women without labor pain relief and the daily activities of the group of women with intravenous anesthesia (*p* = 0.01). Statistically significant associations between the quality of sleep (*p* = 0.275), daily activities (*p* = 0.4) and epidural analgesia was not stated.

## 4. Discussion

In Yerby’s [10] *Pain in Childbearing: Key Issues in Management*, the author noted that 41% of women considered labor as the worst experience they had ever had. These statements prove that, in a modern society, management of pain during delivery is a crucial moment in childbirth aimed not only at providing women with more comfort, but also to relieve their stress and suffering [1,2,3,4]. Epidural analgesia is an effective and widely used treatment for labor pain. Even though the frequency of the use of epidural analgesia varies around the world, an increasingly growing trend is observed in this segment of obstetrics. In Canada, the epidural rate varies among the provinces from 30% to 69%. The use of epidural analgesia in the US has tripled between 1981 and 2001, with 60% of women using this technique in large hospitals [11,12].

The present survey showed that epidural analgesia was more frequently preferred by primiparous women. Rimaitis et al. [13] and Basarinaite et al. [14] published the same results [13,14] revealing that women who arrived for the first labor were more stressed and fearful of labor pain, and therefore, were more likely to opt for epidural analgesia to avoid possible discomfort during delivery.

Much research has been done to assess the influence of epidural analgesia on the mean delivery time. Abbasi et al. [8] reported that the mean duration of labor was significantly higher in women with epidural analgesia compared to groups of non-epidural analgesia women. Our findings revealed the same outcomes. Women with epidural analgesia gave birth on average 37 min longer than women of other groups. According to data from the literature, these results have an important clinical significance for the general condition of newborns. This is reflected by the changes in the blood gas content of newborns. Prolonged labor, and especially the second stage of delivery, directly affect the amount of lactate in the umbilical cord artery: the longer the second stage of labor, the greater the amount of lactate. It shows an abnormal fetal blood flow which results in hypoxemia, hypercapnia, and metabolic acidosis which can lead to long-term complications in the future [15].

The choice of epidural analgesia depends on multiple factors: social and psychological factors, individual desires, and determination. Also, hospital facilities and doctor’s recommendations are very important. Many studies have been conducted to evaluate the factors that influence the decision to choose epidural analgesia during labor [16,17,18]. One of the most important factors was back pain after previous delivery. The current survey showed that the incidence rate of epidural analgesia was significantly lower in women with back pain after previous labor (*p* = 0.009). The same results were obtained by Wong and To [19] who published a study in which 261 women were surveyed before labor and 365 women after labor. The authors concluded that the presence of back pain symptoms after previous delivery was associated with significantly fewer requests or preferences for epidural analgesia in this labor [19]. This evidence shows that despite the fact that the most advanced techniques and the latest medications have been employed, post-partum back pain remains among the largest fears of women leading them to choose the epidural analgesia technique.

Epidural analgesia in labor was first associated with post-partum back pain in 1969, and in 1990, a retrospective study suggested that intrapartum epidural analgesia caused subsequent back pain [20,21]. On the contrary, other studies have revealed that epidural anesthesia was not associated with an increased risk of post-partum back pain [3,6,11,12]. In 2011, Anim-Somuah et al. [3] published the review *Epidural versus Non-Epidural or No Analgesia in Labor*, which included 38 studies involving 9658 women. Three studies included 1806 female participants and assessed the effect of epidural pain relief on post-partum back pain. There was no significant difference observed among these subjects [3]. The Australian study which included 992 women found no statistically reliable relationship between the epidural pain relief method applied and the post-partum back pain [20]. The present survey has confirmed the results of the abovementioned prospective trials: epidural analgesia does not increase the risk of back pain in the early post-partum period and six months after the delivery.

Our study has found that the incidence of post-partum back pain a few days after labor was similar in all research groups (30%). Abbasi et al. [8] also reported similar back pain frequency in the early period after labor (40%) in both groups. The results of the Butler and Fuller [22] studies coincide with the outcomes of our study in assessing the incidence of back pain (30.5%) in both groups during the post-partum period [22]. In the early post-partum period, back pain was probably caused by fatigue and intense physical exertion during labor and postural changes immediately after delivery [20].

The frequency of back pain in the late post-partum period remained similar. It ranged from 23.91% up to 31.65%, but there were no statistically significant differences observed among the groups. Loughnan et al. [23] found 49% of the frequency of back pain 6 months after delivery. On the contrary, Butler and Fuller [22] identified an appropriate epidural analgesia and non-epidural analgesia group of 7.5% and 6.9% back pain frequency 3 months after delivery, respectively. In the later period after labor, the frequency of back pain can be increased by bone–muscle pain due to the newborn’s breastfeeding and general care [20].

The frequency of the accompanying symptoms of back pain varied from 3.23% up to 12.9% a few days after delivery. Six months after delivery, the subjects were surveyed again. The incidence of accompanying symptoms increased to 27.87%; however, there was no significant difference observed among these subjects. Most frequently, women suffered from headaches (22.95%), shoulder (26.23%), and neck (27.87%) pain. MacArthur et al. [24] found that epidural analgesia during delivery can cause various problems in women: headache, migraine, neck pain, tingling in the arms and legs, dizziness, and fainting. The data from this study were obtained by evaluating women’s questionnaires sent to them by post from 13 months up to 9 years after delivery, regardless of the social and obstetric characteristics of women, therefore, the results of this study are not sufficiently statistically reliable. Our study evaluated the abovementioned and other accompanying symptoms and did not observe a statistically significant relationship between the symptoms related to back pain and the pain relief groups. Goldsmith et al. [25] studied the prevalence of post-partum headaches during the three months after delivery and found that as much as 39% (381 women out of 985) of women suffer from headache and neck pain.

## 5. Conclusions

Our survey found that labor pain relief techniques deployed did not increase the risk of back pain in the early post-partum period and six months after delivery. The epidural analgesia was more frequently selected by primiparous women and this pain relief technique increased the mean delivery time as well as the duration of the first and second labor stages.

## Figures and Tables

**Figure 1 medicina-55-00354-f001:**
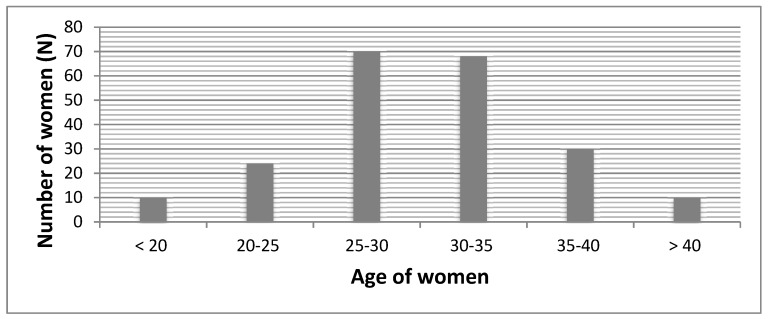
Distribution of women by age groups.

**Figure 2 medicina-55-00354-f002:**
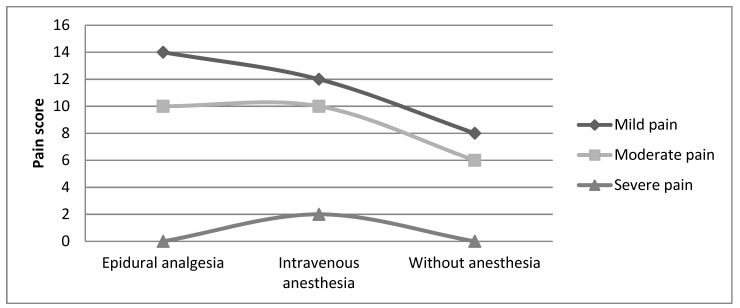
Relationship between types of analgesia and intensity of post-partum back pain.

**Figure 3 medicina-55-00354-f003:**
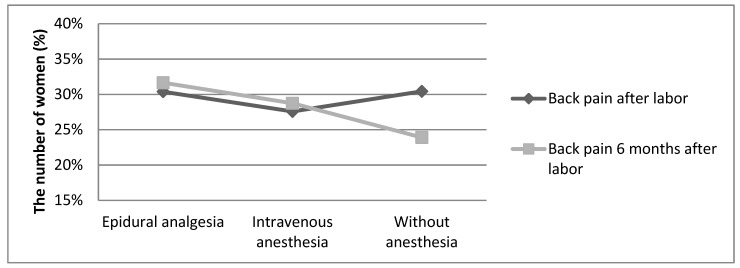
The prevalence of post-partum back pain after delivery and six months after delivery.

**Table 1 medicina-55-00354-t001:** Demographic characteristics of women.

Characteristics	Epidural Analgesia	Intravenous Anesthesia	Without Anesthesia	*p*-Value
Age (years ± SD)	29.8 (±5.0)	28.9 (±6.0)	31.3 (±5.4)	0.070
Weight gain (kg ± SD)	14.2 (±5.2)	14.4 (±5.1)	16.3 (±4.3)	0.059
Height (cm ± SD)	168.1 (±6.9)	167.3 (±6.8)	170,1 (±6.4)	0.071
Education				
Primary (*N* (%))	2 (2.5%)	6 (6.9%)	2 (4.3%)	0.502
Secondary (*N* (%))	16 (20.3%)	22 (25.3%)	8 (17.4%)	
Higher (*N* (%))	61 (77.2%)	59 (67.8%)	36 (78.3%)	
Marital status				0.060
Married (*N* (%))	69 (87.3%)	67 (77.0%)	42 (91.3%)	
Single (*N* (%))	10 (12.7%)	20 (23.0%)	4 (8.7%)	

**Table 2 medicina-55-00354-t002:** Obstetric characteristics of women.

Characteristics	Epidural Analgesia	Intravenous Anesthesia	Without Anesthesia	*p*-Value
Parity				
Primiparous (*N* (%))	48 (60.8%)	42 (48.3%)	10 (21.7%)	0.002
Multiparous (*N* (%))	31 (39.2%)	45 (51.7%)	36 (78.3%)	
Duration of labor (minutes ± SD)	510.1 (±180.6)	491.3 (±210.1)	374.6 (±178.0)	0.001
Duration of first stage of labor (minutes ± SD)	458.1 (±173.5)	446.8 (±194.2)	342.4 (±17.9)	0.002
Duration of second stage of labor (minutes ± SD)	45,1 (±27.9)	35.5 (±21.7)	23.1 (±10.6)	<0.001
Induced labor (*N* (%))	42 (55.3%)	28 (36.8%)	6 (7.9%)	<0.001
Use of oxytocin (*N* (%))	58 (73.4%)	50 (57.5%)	14 (30.4%)	<0.001
Instrumental vaginal delivery (*N* (%))	8 (10.1%)	2 (2.3%)	0 (0%)	0.014
Perineal tear (*N* (%))	26 (27.7%)	40 (42.6%)	28 (29.8%)	0.011
Episiotomy (*N* (%))	54 (49.1%)	40 (36.4%)	16 (14.6%)	<0.001

**Table 3 medicina-55-00354-t003:** Data from previous deliveries.

Characteristics	Epidural Analgesia (*N*)	(%)	Intravenous Anesthesia (*N*)	(%)	Without Anesthesia (*N*)	(%)	*p*-Value
Pain relief method							
Epidural analgesia	25	75.8	11	24.4	4	11.1	<0.001
Intravenous anesthesia	4	12.1	22	48.9	10	27.8	
Without anesthesia	4	12.1	12	26.7	22	61.1	
Back pain after previous labor	12	15.4	4	4.5	6	13.0	0.009

**Table 4 medicina-55-00354-t004:** Accompanying symptoms.

Accompanying Symptoms	Frequency (*N*)	Percentage (%)	*p*-Value
Headache	8	12.9	0.097
Shoulder pain	4	6.5	0.754
Neck pain	4	6.5	0.065
Urinary disorders	6	9.7	0.694
Tingling in the hands	2	3.2	0.113
Tingling in the legs	4	6.5	0.234

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
