# Peer review of "Epidural Analgesia and Back Pain after Labor"

_1010-660X, 2019, doi:10.3390/medicina55070354_

Round 1

Reviewer 1 Report

Introduction.The aim of the study is not clearly presented. It is stated that the survey will assess the influence of epidural analgesia on postpartum back pain in postpartum women without epidural analgesia. The hypothesis of the study is missing.

Materials and methods. Explain what final weight represents. It is not described what measures are used in order to analyse the quality of sleep and daily activities.

Results. Tables 1 and 2. To what comparisson does the p-value refer to? The following statements are not clear:

-There was a significant difference.... (lines 96-97).

- There was no statistically significant correlation.... (lines 119-121)

Table 4. What does the p-value compare?

Discussion. Name the previous studies the authors refer to.

The current study does not present what brings new in the field of epidural analgesia and back pain.

Author Response

 Introduction.The aim of the study is not clearly presented. It is stated that the survey will assess the influence of epidural analgesia on postpartum back pain in postpartum women without epidural analgesia. The hypothesis of the study is missing.

Response to point 1: We clarified the aim of study:  The aim of this survey was to assess the influence of epidural analgesia on postpartum back pain in postpartum women.

Materials and methods. Explain what final weight represents. It is not described what measures are used in order to analyse the quality of sleep and daily activities. 

Response to point 2: We explained what measures are usedThe primary outcome variable was back pain quantified by self reports (yes/no), pain scores (digital analog scale) and the back pain impact on daily activities and the quality of sleep (yes/no).

Results. Tables 1 and 2. To what comparisson does the p-value refer to? The following statements are not clear:

-There was a significant difference.... (lines 96-97). 

Response to point 3: There was a significant difference between the pain relief technique used by primiparous and multiparous (p=0.002). 

- There was no statistically significant correlation.... (lines 119-121)

Response to point 3: We deleted "Table 2"", because this p value is not from this table: There was no statistically significant correlation observed between post-partum back pain and the type of anesthesia (p=0.907) . 

Table 4. What does the p-value compare?

Response to point 4:The comparison of the incidence of various accompanying symptoms among the anesthesia groups was not statistically significant.

Discussion. Name the previous studies the authors refer to. 

Response to point  5: In Yerby M. book “Pain in Childbearing: Key Issues in Management “ has  mentionthat 41% of women considered labor as the worst experience they had ever had [23]. 

The current study does not present what brings new in the field of epidural analgesia and back pain.

Response to point 6: This study is absolutely new in our country and original because is prospective study.

Reviewer 2 Report

The manuscript describes a relatively simple study to evaluate the impact of epidural 18 analgesia on post-partum back pain in post-partum women with and without epidural analgesia.This study was well conducted. 

1) You should unify the %, years, min, kg, and cm  notation to the first decimal place.

 e.g.1) 2 (2.53%) to 2 (2.5%) in Table 1.

 e.g.2) 29.81+-(4.95) to 29.8 +-(5.0) in Table 1.

2) You should add error bar in figures.

3) Statistical power should be assessed in Method.

Author Response

1) You should unify the %, years, min, kg, and cm  notation to the first decimal place.

Response to point 1: We corrected the tables

2) You should add error bar in figures.

Response to point 2: our figures show distribution between number of women. There is no mean value that is why figures do not have error bar.

3) Statistical power should be assessed in Method.

Response to point 3: The primary outcome variable was back pain quantified by self reports (yes/no), pain scores (digital analog scale) and the back pain impact on daily activities and the quality of sleep (yes/no).

The data were processed by using Microsoft Excel®. The statistical analysis of the data was conducted by using SPSS® 23.0 software. The average ± standard deviation was employed to evaluate both quantitative (normal distribution) and qualitative data - frequency/percentages. The relationship between the quantitative variables was evaluated by carrying out a one-factorial dispersion analysis (ANOVA test) with a confidence interval (95% CI). The value of p<0.05 was considered statistically significant.

Medicina EISSN 1010-660X Published by MDPI AG, Basel, Switzerland RSS E-Mail Table of Contents Alert
Back to Top